# Perceptions of Parents towards COVID-19 Vaccination in Children, Aseer Region, Southwestern Saudi Arabia

**DOI:** 10.3390/vaccines10081222

**Published:** 2022-07-30

**Authors:** Ayed A. Shati, Saleh M. Al-Qahtani, Abdullah A. Alsabaani, Syed E. Mahmood, Youssef A. Alqahtani, Khalid M. AlQahtani, Mohammed S. Aldarami, Fahad D. AlAmri, Abdulrahman Saad Alqahtani, Abdulrahman M. AlHadi, Ausaf Ahmad, Fatima A. Riaz

**Affiliations:** 1Department of Child Health, College of Medicine, King Khalid University, Abha 62529, Saudi Arabia or ashati@kku.edu.sa (A.A.S.); smuadi@kku.edu.sa (S.M.A.-Q.); youssef9811@hotmail.com (Y.A.A.); 2Department of Family and Community Medicine, College of Medicine, King Khalid University, Abha 62529, Saudi Arabia; aalsabaani@kku.edu.sa (A.A.A.); fatima.riaz786@yahoo.com (F.A.R.); 3Medical Intern, College of Medicine, King Khalid University, Abha 62529, Saudi Arabia; kmfare@windowslive.com (K.M.A.); msaldarami@gmail.com (M.S.A.); 4Medical Student, College of Medicine, King Khalid University, Abha 62529, Saudi Arabia; dr.fahad.1@hotmail.com (F.D.A.); da7mi.07@gmail.com (A.S.A.); ysyras1999@gmail.com (A.M.A.); 5Department of Community Medicine, Integral University, Kursi Road, Lucknow 226026, India; ausaf.ahmad86@gmail.com

**Keywords:** perception, parents, COVID-19, vaccination, children, Aseer region, Saudi Arabia

## Abstract

Vaccines are an important part of the COVID-19 pandemic response plan. This cross-sectional study aims to assess the attitude and perception levels of parents toward COVID-19 vaccines for children aged 0–18 years in the Aseer region of Saudi Arabia. Data were analyzed using SPSS version 16.0. Out of a total of 1463 parents, 30.6% assumed that COVID-19 vaccination may be more dangerous for children than adults. Nearly 36.5% parents don’t have any concern about children’s vaccination. About 12.8% of children have not received the vaccination, 55% of parents have some sort of hesitation and 32.2% of parents did not hesitate before vaccinating their children against COVID-19. Only 15.4% of parents expect that the COVID-19 vaccine affects their child’s genes. About 23.4% parents strongly agreed and 35.1% agreed about the importance of getting their children vaccinated. About 22.1% of parents strongly agreed and 33.3% agreed regarding their willingness to get their children vaccinated to prevent Coronavirus disease. More than 80% of parents recommended rushing to receive the COVID-19 vaccine. Health professionals and policymakers should implement and support strategies to ensure children are vaccinated for COVID-19. They also need to educate parents and families regarding childhood vaccination.

## 1. Introduction

In Saudi Arabia, as of 22 July 2022, there have been 805,879 confirmed cases of COVID-19 with 9233 deaths, reported to WHO. A total of 66,700,629 vaccine doses have been administered in the entire Kingdom [1]. Vaccines are an important part of the COVID-19 pandemic response plan. They resemble one of the most successful and cost-efficient public health interventions developed, saving millions of lives every year [2,3,4]. Following the decoding of the SARS-CoV-2 genome sequence in early 2020 [5] and the World Health Organization’s (WHO) declaration of the pandemic in March 2020 [6], scientists and pharmaceutical firms are racing against the clock to create effective vaccines [7]. Around the world, there are now 137 COVID-19 vaccine candidates undergoing clinical trials and 194 candidates in pre-clinical development [8]. Four vaccines (AstraZeneca Oxford, Moderna, Pfizer and Johnson & Johnson) have been approved for use in Saudi Arabia [9].

In a cross-sectional study conducted using interviews with parents visiting outpatient clinics at Saudi Arabia, twenty percent of parents were found hesitant about childhood vaccines. In multivariate analyses, the main factors associated with both parents’ hesitancy and incomplete vaccination status were believing that vaccines are ineffective (adjusted odds ratio [AOR] = 28, 95% and believing that vaccines are not important (AOR = 27) [10]. Misinformation steering towards vaccine hesitancy could put public health in harm and cause obstacles in responding to the current crisis [11]. There is a need of disseminating accurate health information regarding COVID-19 to the general public through several mediums (news, social media, and government websites) to rectify misconceptions. The impact of media exposure may be linked to the dissemination of critical pandemic health information. Although early media exposure seems to have promoted health-protective actions, as the epidemic progresses, media fatigue—when individuals grow accustomed to continuous messaging—may diminish this impact [12].

Based on data from death certificates, autopsy reports, mental history, and clinical descriptions from Vaccine Adverse Event Reporting System (VAERS) reports and health care professionals, there is no indication of a causal link between COVID-19 vaccinations and death. Only a few instances of allergy have been recorded after receiving the COVID-19 vaccinations from Pfizer-BioNTech and Moderna (4.5 cases per million doses given) [13]. 

Canada became the first nation in the world to approve the COVID-19 vaccine for emergency use in children aged 12–15 years on 5 May 2021; later that month, the US Food and Drug Administration and the European Medicines Agency approved the Pfizer-BioNTech COVID-19 vaccine for adolescents [14]. Children under the age of 12 years are the next group to benefit from a COVID-19 vaccination that is both safe and effective. Bihua Han et al. in their double-blinded, randomized, controlled study found that the inactivated COVID-19 vaccine (CoronaVac) had excellent safety, tolerability, and immunogenicity in children aged 3–17 years [15]. This finding should encourage researchers to continue testing additional COVID-19 vaccinations in children under the age of 12 years. Children accounted for 14.1% of all COVID-19 cases in the United States [16]. COVID-19 is typically mild and asymptomatic in children. Children may, however, get severely sick and need hospitalization and urgent treatment in rare instances. Multisystem inflammatory syndrome in children (MIS-C) is one of the potential negative consequences; children with MIS-C have multisystem diseases affecting the heart, lungs, kidneys, brain, skin, eyes, and gastrointestinal tract after developing a fever and becoming highly inflamed. The total mortality rate with MIS-C is about 1–2% [17]. Because the BNT162b2mRNA (COVID-19) vaccine has demonstrated 100% effectiveness in children aged 12–15 years, these negative consequences of COVID-19 in children supported the need to vaccinate children against COVID-19. Parents have reported differing opinions on frequencies and risks of coronavirus disease transmission and medical complications and of effectiveness and adverse effects of a vaccine in the Jeddah and Riyadh regions of Saudi Arabia [18]. Vaccine hesitancy is prevalent among parents of 5–11-year-old children in Saudi Arabia and those who have beliefs of minimal benefits or lack of safety from the COVID-19 vaccine are more hesitant [19].

The studies on this particular issue are scarce and to the best of our knowledge no study has been reported from Aseer region till date especially taking into account the age group between 0–18 years. The study findings would fill the gap in the literature. Therefore we have made an attempt to assess the parent’s attitude and perception regarding vaccine acceptance for children and to frame specific strategic interventions to create a positive perception of the COVID-19 vaccine. This study aims to assess the attitude and perception levels of parents toward COVID-19 vaccines for children sin the Aseer region of Saudi Arabia.

## 2. Materials and Methods

### 2.1. Study Design &Duration 

This cross-sectional study was conducted from 1 August 2021 to 15 March 2022.This study was approved by the Research Ethical Committee of the College of Medicine, King Khalid University (REC#2021).

### 2.2. Population and Sample Size

The study included both Saudi and Non–Saudi nationals residing in Aseer region, Saudi Arabia. We included male and female parents who were currently having children aged between 0 and 18 years and also those not having children. 

Moreover, assuming the maximum variability, which is equal to 50% (*p* = 0.5) and taking 95% confidence level with ±5% relative precision, the calculation for required sample size will be as follows: So, using the formula *n* = z2pq/(pl)2, And putting in values as, *p* = 0.5 and hence q = 1 − 0.5 = 0.5; l = 0.05; z =1.96, 

n =(1.96)2(0.5)(0.5)/(0.5 × 0.05)^2^n = 1537

We removed 74 samples from surveys as these responses do not represent genuine preferences such as incomplete information, or wrong information etc. So, statistical analysis and results were done on a 1463 sample. 

### 2.3. Data Collection

Data was collected using an adapted and modified questionnaire from the study of El-Elimat et al. [17]. The questionnaire was distributed online to the parents and comprised of closed ended questions made to meet the study objectives. The online questionnaire was prepared in both English and Arabic language using Google forms and distributed among participants through social media and E-mail for convenience of data collection as face-to-face interviews had to be avoided following the social distancing norms by the government. The questionnaire was translated from English to Arabic (local language) by a bilingual person to enable an easy understanding of the questions and avoid any questionnaire bias. Before administration of the final version of the questionnaire, a pretest was performed among random parents in the region to ensure the reliability and applicability of the questionnaire. The results of the piloted study were not included in the final analysis. The questionnaire comprised of three sections; the first included the participants’ sociodemographic characteristics, such as age, sex, and nationality. The second part included questions assessing the attitude of parents towards COVID-19 vaccines for children. The third part consisted of questions assessing the perspectives of parents towards COVID-19 vaccines for children.

### 2.4. Ethical Considerations

The questionnaire started with a brief explanation of its objective and intent and a reminder to participants that their participation is entirely voluntary. The survey didn’t collect names, nor will they collect dates of birth or addresses. Electronic informed consent was obtained from all the participants before filling out the survey forms. The confidentiality of data was well-preserved throughout the study by keeping it anonymous and asking the participants to select honest answers and options.

### 2.5. Statistical Analysis

The collected data were coded and entered into an excel software (Microsoft office Excel 2010) database. Data was analyzed using Statistical Package for Social Sciences, version 16.0 (SPSS, Inc., Chicago, IL, USA). Information related to parent’s perspectives and their experience towards COVID-19 vaccine were presented in descriptive statistics like, frequency and percentage. Pearson’s chi-square test was used to assess the association between perception of parent’s expectation that vaccination may be more dangerous for children than adults as a dependent variable and related risk factors such as socio-demographic, queries related to parents’ perspectives variables as independent variables. *p*-value of less than 0.05 was regarded as statistically significant.

## 3. Results

Table 1 shows the association between parents’ socio-demographic variables and their concern towards children’s vaccination. A total of 1463 parents were included in the study. In which 449 (30.69%) assumed that vaccination may be more dangerous for children than adults. Out of 449 study subjects, the frequency of female parents was higher as compared to male parents. Approximately one third of parents expect that vaccination may not be more dangerous for children than adults. The majority of parents 534 (36.50%) don’t have any concern about children’s vaccination. In the age group 20–30 years, more than 205 (40.8%) of parents think that vaccination may not be more dangerous for children than adults and 160 (31.8%) expect that it may be more dangerous for children than adults. The majority of parents whose education levels were above bachelor’s degree and those belonging to the medical field perceived that vaccination may not be more dangerous for children than adults. Most of the parents with one child think that vaccination may not be more dangerous for children than adults 224 (42.6%) followed by parents with two children 149 (28.7%). Statistical significant differences were found between parents’ sociodemographic variables such as gender, age, marital status, occupation, employment, nationality, monthly income, having children, and the number of children with their concern towards childhood vaccination. Table 2, Figure 1 and Figure 2 shows the distribution of parents’ standpoints on the Coronavirus (COVID-19) Vaccination. It was observed that 12.8% of parents have not concern for vaccination, 805(55%) of parents had some sort of hesitation and 471(32.2%) of parents did not hesitate before vaccinating their children. The majority of parents strongly agreed 342(23.4%) and agreed 514(35.1%) about the importance of getting their children vaccinated. Parents strongly agreed 323 (22.1%) and agreed 487 (33.3%) regarding their willingness to get their children vaccinated to prevent Coronavirus disease. The majority of parents do not agree 387(26.5%) that fears prevent them from administering the vaccine to protect their children from the Coronavirus due to the side effects. Most parents do not agree 475 (32.5%) and very few 125(8.5%) strongly agreed that the COVID-19 vaccine could affect their children’s puberty or fertility. In Figure 1 yellow represent that the 48% parents don’t know about whether corona vaccine affects child genes or not. More than 80% of parents recommended rushing to receive the Corona vaccine. The majority of parents 1030(70.4%) opined that the services provided at the vaccination centres were excellent. Figure 2 shows that the most reliable sources of information for parents about vaccines were health care providers (26.6%) followed by the Food and Drug General Organization (25.2%). Another important source of information were social media (facebook, twitter, whatsapp) (21.8%) and media (tv, radio) (17.4%).

Table 3 shows the frequency distribution of parents experience with coronavirus. About 847(57.9%) of parents had not experienced any bad side effects from receiving vaccinations in the past. About 1041 (71.2%) of parents did not get the seasonal flu shot last year. Nearly 293(20.0%) of the respondent’s family members, followed by respondents themselves 180(12.3%) and the respondent’s friends 109 (7.5%) had been confirmed to be infected with the Coronavirus through laboratory examination. It was also found that 971 (66.4%) of parents did not think that they may have been exposed to or infected with Coronavirus (without testing) (Table 3).

Figure 3 shows that the nearly 30.7% thought that vaccination may be more dangerous for children than adults, whereas 32.8% of parents admitted that vaccination may not be more dangerous for children than adults. In Figure 3 yellow represent that the 36% parent’s don’t have concern about vaccination being more dangerous for children than adults. Figure 4 shows that the fear of infection by a family member (13.0%) was the commonest worry followed by fear of contracting the virus (7.6%) in the parents. However, 11.8% of parents were not worried about any problem.

Figure 5 depicts that the parents’ attitudes towards vaccination, immunity, and the coronavirus (COVID-19) pandemic. In which majority of parents strongly agreed 306 (20.9%) and agreed 541 (37.0%) that vaccines are generally considered safe.

Parents who believe that COVID-19 vaccine is more dangerous for children than adults among them almost half of them 225(47.8%) showed hesitancy for their children being vaccinated, 126 (56%) have concerns that Corona vaccine affects child genes, majority of them 57 (60%) were not willing to give the vaccine to prevent Coronavirus disease to your children, half of them 65 (52%) showed concerns that the COVID-19 vaccine could affect your son/daughter’s puberty or fertility and 167 (45.6%) had any bad side effects from receiving vaccinations in the past whereas statistical significance differences were found between parents who believe that COVID-19 vaccine is more dangerous for children than adults with the hesitancy before your children being vaccinated, have concerns that Corona vaccine affects child genes, willing to give the vaccine to prevent Coronavirus disease to your children, have concerns that the COVID-19vaccine could affect your son/daughter’s puberty or fertility and who had any bad side effects from receiving vaccinations in the past (Table 4).

Table 5 shows that the queries related to parents’ perspectives as response in strongly agree/agree strongly disagree/I do not agree. Eight hundred ten parents strongly agreed/agreed for query ‘willing to give the vaccine to prevent Coronavirus disease to your children’. While, only 180 parents expect that vaccination may be more dangerous for children than adults. Three hundred twenty-three parents strongly agreed/agreed about that COVID-19 vaccine could affect your son/daughter’s puberty or fertility. Results show statistically significant association.

## 4. Discussion

Vaccination is one of the most useful scientific interventions which have been proved for controlling many infectious diseases and has helped to eradicate some diseases. Our aim of the study was to identify factors driven towards COVID vaccine hesitancy or acceptance for children among parents. One of the major factors towards any vaccine acceptance and hesitancy is the parental perception of that vaccine [20]. The report of the World Health Organization (WHO) regarding immunization revealed that in 2020, the number of unvaccinated children increased by 3.4 million as compared to the number of children who received vaccination during 2019 which was showing a decline in the total percentage from 86% to 83%. Certain misconceptions are being circulated about different vaccines although studies are being done to find the reasons for this decline [21]. 

We are observing similar problems related to myths and beliefs in the general population regarding acceptance of COVID-19 vaccination not only for children but for adults as well. Therefore, it is necessary to quantify the extent of the problem and to identify measures to reduce the COVID-19 vaccination hesitancy among parents and children [22]. 

COVID-19 vaccine is the most powerful tool that we have in combating the battle against the COVID-19 pandemic, but we found an increased rate of vaccine refusal and scepticism toward the vaccine. A recently published report demonstrated that four out of ten Americans will definitely or probably not take the vaccine and about 21% of American adults are “pretty certain” that they will not take the vaccine. The increasing rate of vaccine refusal or hesitancy is driven by diverse factors in different countries or community groups [23]. 

Our online survey targeted those parents who were interested in participating in the study and provided a relevant snapshot of the COVID-19 vaccine perspective of parents. Mei-Xian Zhang et al. demonstrated quite many similarities in their study, female parents were more responsive to questionnaires, and female parents were having more concerns about vaccine safety, although they reported 46% willingness of parents to vaccinate their children whereas in our study 55% parents (22% parents strongly agree and 33% parents agreed) for vaccination for their children [24]. A higher proportion of highly educated Saudi parents thought that vaccine is not dangerous for children in our study. Other researchers have also reported that higher educational levels were associated with higher confidence toward vaccination [25,26,27]. 

Around 40% of parents in the age group 20–30 years in our study believed that vaccine is not dangerous for children indicating that the young Saudi generation is more educated than the older age group and hence has more information and knowledge about vaccination and its effects. 

Similar to our study findings, Mei-Xian Zhang et al. also reported that mothers were having more fears than fathers regarding the dangers of the COVID vaccine for their children [2,24].

Similarly, Diego Urrunaga-Pastor et al. demonstrated that parents with one child have more attitude toward getting vaccination similarly our study also showed parents with one child showing a good attitude towards COVID vaccine and having less fear of vaccine than parents with two children probably parents with one child are young and more educated as compared with the parents having more children, as the young generation is more educated in Saudi Arabia than older generation here [2,28]. Multiparty is one of the known factors for delayed immunization or avoiding immunization for young children [2,29].

There are many reasons behind the parental concerns about vaccine safety and its side effects. Many studies have showed concerns not only for the COVID-19 vaccine but Measles, mumps, and rubella (MMR) vaccine and its relation to autism has also played a significant role in vaccine hesitancy and refusal in the past [30,31]. People have some similar beliefs and concerns regarding the COVID-19 vaccine as well because of undefined safety related to the COVID vaccine and certain myths related to vaccine-associated infertility as well [3,32]. Our study findings are also in agreement to this, as more than 50% of parents had concerns that the COVID-19 vaccine could affect their children’s genes, puberty or fertility. 

The US surveillance data reported cases of the multisystem inflammatory syndrome (MIS-C) after at least one dose of COVID-19 vaccine among children under 19 years of age. Yousaf and colleagues identified 21 individuals (median age 16 years, range 12–20) with MIS-C after COVID-19 vaccination. Reports of MIS-C after COVID-19 vaccination occurred in only 1 per million individuals aged 12–20 years who received one or more doses of a COVID-19 vaccine, and 15 (71%) of 21 individuals with MIS-C had laboratory evidence of antecedent SARS-CoV-2 infection, casting doubt about attribution [33,34].

Although 55% of parents showed hesitancy before their children’s vaccination in this study. 

Around 88% of children in our study population were vaccinated because it is mandatory by the Ministry of health of Saudi Arabia that all kids will be vaccinatedfor COVID therefore the rate of vaccination in the study region of Saudi Arabia is higher than other parts of the world. Alfieri et al. reported only 33% vaccination rate in US [35]. 

In our study majority of parents strongly agreed (23.4%) and agreed (35.1%) about the importance of getting their children vaccinated, 22.1% of parents strongly agreed and (33.3%) agreed regarding their willingness to get their children vaccinated to prevent Coronavirus disease. Marco Montalti et al. also reported that about 60% of the guardians of children under 19 years of the Metropolitan City of Bologna are also inclined to vaccinate their children [3,27].

About 26% of parents did not agree to vaccinate their children against COVID-19 because of the fear of side effects whereas the majority of them were thinking vice versa.

Only 15.1% of parents believe that the COVID-19 vaccine can effect the genes of their children. This is probably as they were mistaken with the type of vaccines like the Messenger RNA (mRNA) vaccine and the technique which is known as reverse genetics (RG) that is widely used for the genetic manipulation of RNA viruses from their full-length cloned DNA (cDNA) and can lead to the development of successful counter measures [3,36].

Nearly 8.5% and 13.5% of parents respectively strongly agree and agree that vaccines can affect the puberty and fertility of their children. Malik Sallam et al. also reported that 23% of the Kuwaiti and Jordanian population believe that vaccines can cause infertility [3,37].

Studies are still insufficient in number regarding the long-term side effects of the COVID-19 vaccine among human beings. Although studies conducted on rats have showed no adverse effects of BNT162b2 (mRNA COVID-19 vaccine) on female rat fertility and reproduction [3,38].

The most reliable sources of information for parents in our study about vaccines were health care providers (25.5%) followed by the Food and Drug General Organization (25.2%) although the most common source of information about COVID-19 vaccines was reliance on medical doctors, scientists, and scientific journals (36.4%) among Jordanian and Kuwaiti population [38].

In contrast, mass media communication and social media platforms are also sources of information but they are associated with more doubts, myths, and disbeliefs regarding the COVID vaccine. This might be linked to the easier spread of misinformation through social media, including inaccurate and wrong information regarding the safety of COVID-19 vaccines [4,39]. 

More than 80% of parents in the Aseer region of Saudi Arabia recommended rushing to receive the COVID-19 vaccine, indicating good health-seeking behaviour in the study population. There is also an urgent need by the administrations and public health agencies to seek political support for full coverage of the COVID-19 vaccination not only in Saudi Arabia but also worldwide [4,40]. 

The majority of parents (57.9%) agreed (strongly agreed 20.9% and agreed 37.0%) that the vaccines are generally considered safe in our study. Katherine Kricorian reported a 41% positive response regarding vaccine safety among the American population [4,41].

Around 20% of respondents’ family members have been affected by COVID-19, whereas 13% of respondents were having fear of COVID infection in their family members, which is lower than that reported (34%) by Liliana Cori et al. [4,42]. Previous research explored the various determinants of vaccine hesitancy among parents and showed that parents who did not have any plan to vaccinate their children for the COVID-19 vaccine had significantly higher levels of misconceptions and mistrust for the COVID-19 vaccines, they also have lower levels of confidence regarding the COVID-19 pediatric vaccine safety and efficacy [4,43].

Previous studies have stated that the safety and efficacy of the pediatric COVID-19 vaccine influences parents’ hesitancy toward pediatric COVID-19 vaccine acceptance by the parents by 19.5% and influences their plan to get their children vaccinated. Whereas in our study parents were showing hesitancy in 20.2% who were thinking that vaccine is more dangerous for children than adults [4,44]. Almalki OS et al. also concluded that parents who perceived low benefit from the vaccine (OR = 16.3; 95% CI, 12.1–21.9) or who had safety or efficacy concerns (OR = 3.76; 95% CI, 3.10–4.58) were among the most hesitant to vaccinate their children [1,19].

In our study, about 52.9% of parents were not thinking that vaccine is more dangerous for children than adults. Similarly, another study showed that 59.8% of parents stated that the vaccine would protect their children from COVID-19 and its complications that’s why they were willing to vaccinate their children [4,45].

There are certain other myths reported related to children’s vaccination for COVID-19 that this vaccine can cause autism (15.2%), mistrust of big pharmaceutical companies (54.2%), and coronavirus has been developed by the government (48.6%) [4,46].

In a worldwide survey, nearly 37.7% were afraid of adverse effects of the COVID vaccine and 5.6% opined that COVID-19 is a self-limiting disease so there is no need for vaccination [47].In a recent students survey in Egypt, about 22.88% believed that vaccines have some long-term genetic side effects [48].Overall female gender is reported to bemore hesitant toward COVID-19 vaccination worldwide [49].Most likely they are worried about puberty and fertility in the future. A massive project conducted across 50 statesalso reported that 37% of the population believe that vaccines have health effects specific to girls and women, and 34% believe that vaccines have health effects specific to boys and men [50].

Parris Diaz et al. also reported that about 41% of adults believed that the COVID-19 vaccines can negatively impact their reproductive health and or fertility, and about 38% of respondents were unsure of the effects on fertility [51]. The above findings are similar to what has been reported by our study. Less than one-half of US participants in a National survey reported that they are likely to have their child receive a COVID-19 vaccine [52]. Majority of parents were going to vaccinate their children against COVID-19 in a previous study in a Saudi population [53]. The prevalent level of hesitancy, fear, and other health concerns observed among the parents towards their children’s vaccination should be reduced. The following factors may have led to certain limitations in the present study. The cross-sectional nature of this study cannot confirm the causality association between the compared variables. The self-reported responses could over or underestimate the results. The required sample size according to sample size calculation was 1537. However, the final sample size was 1463 after removal of 74 samples. This is below the required target and had to be compensated by collection of more samples which could not be done due to time and resource constraints. This study was conducted at one center in Saudi Arabia and our findings may not present parents’ view regarding COVID-19 vaccine across Saudi Arabia.

We hope in the future to have all the required resources to do multicentric/nationwide studies However the data included children of all age groups between 0–18 years and an extensive analysis has been made is the strength of our study. This study assessed the parent’s attitude and perception regarding vaccine acceptance for children in the study population. We learnt that, 30.6% parents assumed that COVID-19 vaccination may be more dangerous for children than adults. About 12.8% of children have not received the vaccination and 55% of parents have some sort of hesitation before vaccination. Nearly 15.4% of parents expect that the COVID-19 vaccine affects their child’s genes.

## 5. Conclusions

Thus our study shows significant differences between parents who believe that the COVID-19 vaccine is more dangerous for children than adults with the hesitancy before your children being vaccinated, have concerns that COVID-19 vaccine affects child genes, willing to give the vaccine to prevent Coronavirus disease to your children and have concerns that the COVID-19 vaccine could affect your son/daughter’s puberty or fertility. Health professionals and Policymakers should implement and support strategies to ensure children are vaccinated for COVID-19. They also need to educate parents and families regarding importance of vaccination against COVID-19. A support program for health education for these parents should be introduced. This can go a long way to reduce the impact of hesitancy, fear, among them. The study might be helpful in designing interventions to address the impediments in the current COVID-19 vaccination drives and vaccinate a wider child population to limit the pandemic.

## Figures and Tables

**Figure 1 vaccines-10-01222-f001:**
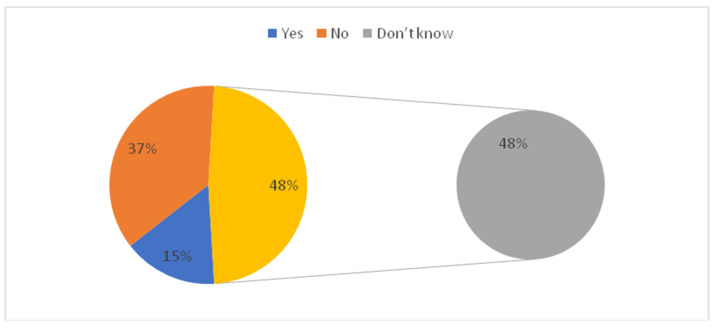
Parents expect that the COVID-19 vaccine affects child genes?

**Figure 2 vaccines-10-01222-f002:**
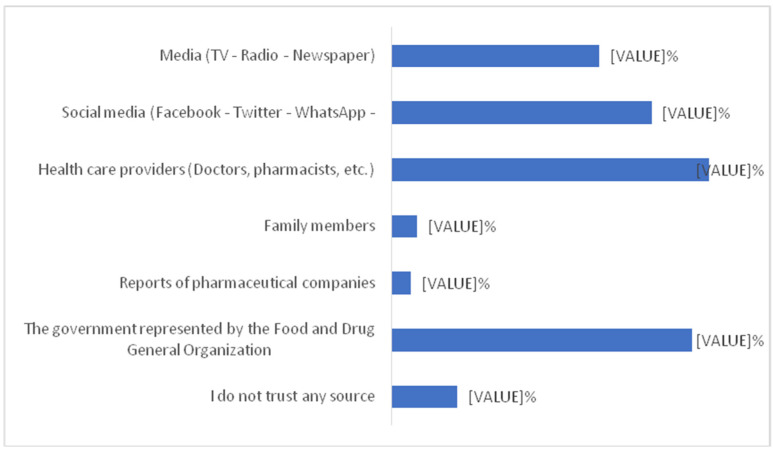
Most reliable sources of information for parents to obtain information about vaccines.

**Figure 3 vaccines-10-01222-f003:**
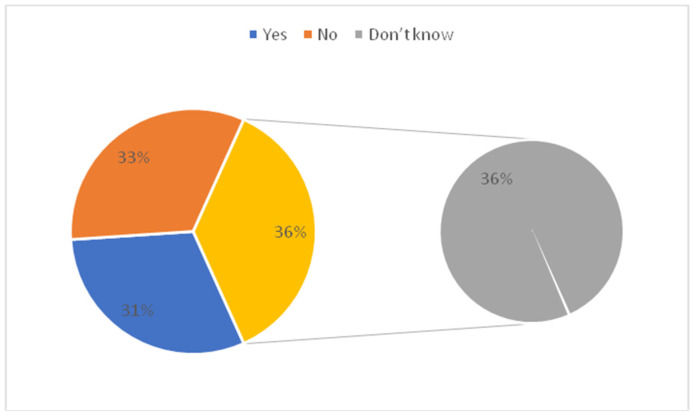
Parent’s attitude about vaccination being more dangerous for children than adults.

**Figure 4 vaccines-10-01222-f004:**
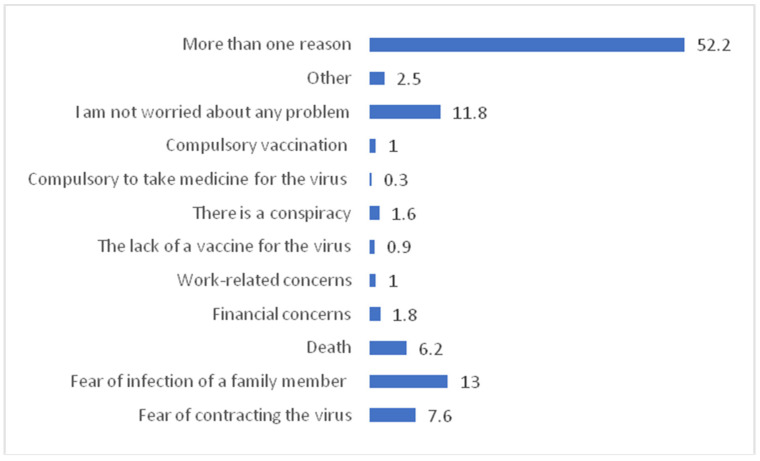
What worries parents faced the most during the COVID-19 pandemic?

**Figure 5 vaccines-10-01222-f005:**
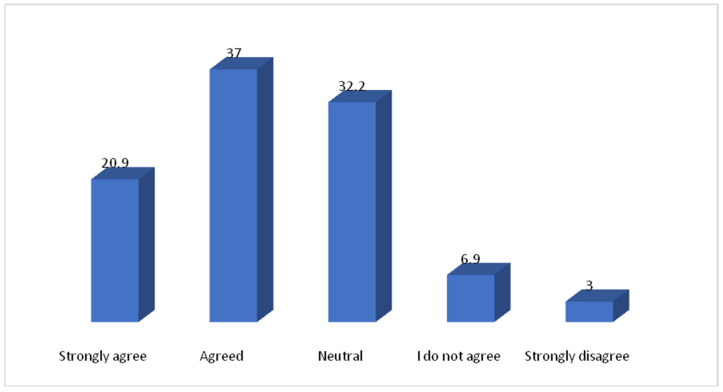
Parents attitude: Is vaccines are generally considered safe?

**Table 1 vaccines-10-01222-t001:** Association between sociodemographic variables and parent’s concern toward children’s vaccination.

Parent’s Sociodemographic Variables	Categories	Parents Expect that COVID-19 Vaccination May Be More Dangerous for Children than for Adults						*p*-Value
		Yes (n = 449)		No (n = 480)		Don’t Know (n = 534)		
		N	%	N	%	N	%	
Gender	Male (n = 623)	148	23.8%	247	39.6%	228	36.6%	0.00
	Female (n = 840)	301	35.8%	233	27.7%	306	36.4%	
Age in years	20–30 (n = 503)	160	31.8%	205	40.8%	138	27.4%	0.00
	31–40 (n = 366)	125	34.2%	105	28.7%	136	37.2%	
	41–50 (n = 433)	135	31.2%	115	26.6%	183	42.3%	
	>50 (n = 161)	29	18.01%	55	34.16%	77	47.83%	
Marital status	Married (n = 1257)	396	31.5%	396	31.5%	465	37.0%	0.02
	Divorced (n = 102)	33	32.4%	38	37.3%	31	30.4%	
	Widowed (n = 104)	20	19.2%	46	44.2%	38	36.5%	
Education	Uneducated (n = 18)	6	33.3%	8	44.4%	4	22.2%	0.22
	Primary school (n = 413)	121	29.3%	121	29.3%	171	41.4%	
	Bachelor’s degree (n = 890)	283	31.8%	292	32.8%	315	35.4%	
	Master’s degree (n = 142)	39	27.4%	59	41.5%	44	30.9%	
Occupation	Medical specialty (n = 292)	82	28.1%	137	46.9%	73	25.0%	0.00
	Non-medical specialty (n = 1171)	367	31.3%	343	29.3%	461	39.4%	
Employment	Employed (n = 826)	253	30.6%	277	33.5%	296	35.8%	0.00
	Retired (n = 145)	27	18.6%	43	29.7%	75	51.7%	
	Unemployed (n = 492)	169	34.3%	160	32.5%	163	33.1%	
Nationality	Saudi (n = 1391)	440	31.6%	444	31.9%	507	36.4%	0.00
	Non-Saudi (n = 72)	9	12.5%	36	50.0%	27	37.5%	
Monthly income	Less than 5 thousand riyals (n = 430)	152	35.3%	137	31.9%	141	32.8%	0.00
	From 6–15 thousand riyals (n = 725)	211	29.1%	220	30.3%	294	40.6%	
	More than 15 thousand riyals (n = 308)	86	27.9%	123	39.9%	99	32.1%	
Suffering from any chronic diseases	Yes (n = 243)	80	32.9%	65	26.7%	98	40.3%	0.08
	No (n = 1220)	369	30.2%	415	34.0%	436	35.7%	
Having Children	Yes (n = 937)	300	32.0%	256	27.3%	381	40.7%	0.00
	No (n = 526)	149	28.3%	224	42.6%	153	29.1%	
Number of Children	≤2 (n = 1222)	379	31.0%	415	33.9%	428	35.0%	0.00
	>2 (n = 241)	70	29.0%	65	26.9%	106	43.9%	

**Table 2 vaccines-10-01222-t002:** Parent’s Perspectives on the COVID-19 Vaccine.

Queries Related to Parents’ Perspectives	Categories	Frequency	Percent
Hesitancy before your children received COVID-19 Vaccine	Yes	471	32.2
	No	805	55.0
	Not received	187	12.8
Do you think it is important to get the COVID-19 vaccine for your sons/daughters?	Strongly agree	342	23.4
	Agreed	514	35.1
	Neutral	416	28.4
	Don’t agree	120	8.2
	Strongly disagree	71	4.9
	Don’t know	0	0.0
Willing to give the vaccine to prevent Coronavirus disease to your children	Strongly agree	323	22.1
	Agreed	487	33.3
	Neutral	340	23.2
	I do not agree	95	6.5
	Strongly disagree	55	3.8
	Don’t know	163	11.1
Fears prevent you from administering the vaccine to protect your sons/daughters from the Coronavirus due to the side effects	Strongly agree	129	8.8
	Agreed	201	13.7
	Neutral	376	25.7
	I do not agree	387	26.5
	Strongly disagree	180	12.3
	Don’t know	190	13.0
Have concerns that the COVID-19 vaccine could affect your son/daughter’s puberty or fertility	Strongly agree	125	8.5
	Agreed	198	13.5
	Neutral	514	35.1
	I do not agree	475	32.5
	Strongly disagree	151	10.3
	Don’t know	0	0.0
Do you recommend rushing to receive the COVID-19 vaccine?	Yes	1193	81.5
	No	270	18.5
How satisfied are you with the services provided at the vaccination centers?	Excellent	1030	70.4
	Very good	270	18.5
	Good	112	7.7
	Acceptable	36	2.5
	Not satisfied	15	1.0

**Table 3 vaccines-10-01222-t003:** Parents experience towards vaccinations, immunity, and the coronavirus (COVID-19) pandemic.

Parents Experience with Coronavirus	Categories	Frequency	Percent
Have you had any bad side effects from receiving vaccinationsin the past?	Yes	366	25.0
	No	847	57.9
	Not sure	250	17.1
Did you get the seasonal flu shot last year?	Yes	422	28.8
	No	1041	71.2
Which of the following has been confirmed to be infected with the Coronavirus through laboratory examination?	Me personally	180	12.3
	A member of my family	293	20.0
	One of my friends	109	7.5
	One of my co-workers	86	5.9
	One of my neighbors	55	3.8
	Nobody	222	15.2
	More than one laboratory examination	518	35.4
Do you think you may have been exposed to or infected with Coronavirus (without testing)?	Yes	492	33.6

**Table 4 vaccines-10-01222-t004:** Parent’s Perspectives on the COVID-19 Vaccine and their concern toward children’s vaccination.

Queries Related to Parents’ Perspectives	Categories	Parents Expect that Vaccination May Be More Dangerous for Children than Adults						*p*-Value
		Yes (n = 449)		No (n = 480)		Don’t Know (n = 534)		
		N	%	N	%	N	%	
Hesitancy before your children are vaccinated	Yes	225	47.8%	91	19.3%	155	32.9%	0.00
	No	163	20.2%	331	41.1%	311	38.6%	
	Not Received vaccine	61	32.6%	58	31.0%	68	36.4%	
Have concerns that the Corona vaccine affects child genes	Yes	126	56.0%	48	21.3%	51	22.7%	0.00
	No	129	24.2%	295	55.2%	110	20.6%	
	Don’t know	194	27.6%	137	19.5%	373	53.0%	
Have you had any bad side effects from receiving vaccinations in the past?	Yes	167	45.6%	102	27.9%	97	26.5%	0.00
	No	220	26.0%	333	39.3%	294	34.7%	
	Not sure	62	24.8%	45	18.0%	143	57.2%	

**Table 5 vaccines-10-01222-t005:** Parent’s Perspectives on the COVID-19 Vaccine and their concern toward children’s vaccination. (response in Likert scale).

Queries Related to Parents’ Perspectives (Response in Likert Scale)	Categories	Parents Expect that Vaccination May Be More Dangerous for Children than Adults			*p*-Value
		Yes	No	Don’t Know	
Willing to give the vaccine to prevent Coronavirus disease to your children	Strongly agree/Agree	180	338	292	0.00
	Strongly disagree/I do not agree	100	23	27	
Have concerns that the COVID-19 vaccine could affect your son/daughter’s puberty or fertility	Strongly agree/Agree	157	76	90	0.00
	Strongly disagree/I do not agree	140	285	201	

## Data Availability

Not applicable.

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
