# Peer review of "Perceptions of Parents towards COVID-19 Vaccination in Children, Aseer Region, Southwestern Saudi Arabia"

_vaccines, 2022, doi:10.3390/vaccines10081222_

Round 1
Reviewer 1 Report
This paper presents a parental perspective on COVID-19 Vaccination in children.
It is an interesting paper, but needs to be revised to make the objectives and results more understandable.
Abstract
No modifications are necessary.
Introduction
Please specify the four vaccines approved for use in Saudi Arabia
Materials and Methods
Please describe the specific questions of the questionnaire in this session.
Please indicate which variables were used and how they were analyzed in the statistical analysis.
Results
What do the yellow answers in Figures 1 and 3 represent? What is the difference between them and the gray areas?
Discussion
Please provide a summary of what you wanted to find out in this study and what you learned.
Author Response
As suggested by the respected reviewer the following points have been included:
Q. Please specify the four vaccines approved for use in Saudi Arabia
Reply: Four vaccines (AstraZeneca Oxford, Moderna, , Pfizer and Johnson & Johnson) have been approved for use in Saudi Arabia
Q: Please indicate which variables were used and how they were analyzed in the statistical analysis.
Reply: The collected data were coded and entered into an excel software (Microsoft office Excel 2010) database. Data was analyzed using Statistical Package for Social Sciences, version 16.0 (SPSS, Inc., Chicago, IL, USA). Information related to parent’s perspectives and their experience towards COVID-19 vaccine were presented in descriptive statistics like, frequency and percentage. Pearson’s chi-square test was used to assess the association between perception of parent’s expectation that vaccination may be more dangerous for children than adults as a dependent variable and related risk factors such as socio-demographic, queries related to parents' perspectives variables as independent variables. P-value of less than 0.05 was regarded as statistically significant.
Q: What do the yellow answers in Figures 1 and 3 represent? What is the difference between them and the gray areas
Reply: In figure 1 yellow represents that 48% parents don’t know about whether corona vaccine affects child genes or not. In figure 3 yellow represents that 36% parent’s don’t have concern about vaccination being more dangerous for children than adults. Both yellow answer and gray area are representing the same category. We highlighted the majority of parents responses in grey area.
Q. Please provide a summary of what you wanted to find out in this study and what you learned.
Reply: This study assessed the parent's attitude and perception regarding vaccine acceptance for children in the study population. We learnt that, 30.6% parents assumed that COVID-19 vaccination may be more dangerous for children than adults. About 12.8% of children have not received the vaccination and 55% of parents have some sort of hesitation before vaccination. Nearly 15.4% of parents expect that the COVID-19 vaccine affects their child's genes.
Reviewer 2 Report
The article presented for review concerns an interesting and important phenomenon of parents' attitudes towards vaccinating children against COVID. In the context of the returning successive waves of this pandemic, this is an important topic.
The introduction should contain data on the number of COVID-19 cases, deaths and vaccinations in the entire population of the country, which would help to better understand the attitudes of parents.
The description of the test procedure is correct. The study used a fairly simple test to measure parents' perception of children's immunization against COVID. The statistical analysis is correct, but the study is dominated by the analysis of differences (although the analyzes of differentiation due to sociodemographic factors were made for only one question concerning parent's concern toward children's vaccination), and there is a lack of correlation analyzes.
The part presenting the results of the research is complicated and difficult to analyze. In the first part there is a long description of all the results, followed by tables and charts (why does the "I don't know" category appear twice in the pie charts?). Putting a description of the results after each table and graph would help in the perception of the content.
Author Response
Q. The introduction should contain data on the number of COVID-19 cases, deaths and vaccinations in the entire population of the country, which would help to better understand the attitudes of parents.
Ans: As suggested by the respected reviewer the introduction has been revised. Reference 1 is added
Q: Table 1 and 4: reform the columns’ and add separate the precedence’s (%)
Reply: Table 1 and 4 modified as per instruction.
Q: Figure 2 and 4: connect the sources of information accordingly: (scientific and health care providers) and the sources (social media and internet).
Reply: Figures modified as per instruction.
Q: Table 4: suggested better statistical analysis in Linkert scale (for example use in two tables analysis as one variable the participants were answered agree and strongly agree against the participants were answered I do not agree and strongly disagree).
Reply: Table 4 modified as per instruction and separate table 5 constructed for Likert scale .
Please see attachment for further details

Reviewer 3 Report
The paper is interesting and deal with the emerging topic. The introduction is clear and well arranged. The methodology sounds good but the statistical analyses mast be improved. The discussion is good even could be improved.
· Line 20 affiliations 6-10 please merge in one affiliation
· Line 41 (key words) suggested to authors to reform the key words and exclude the numbers.
· Line 55, “In a cross-sectional study conducted using interviews with parents visiting outpatient clinics at King Khalid University Hospital, Riyadh, Kingdom of Saudi Arabia”. Suggested to authors to reform accordingly: “In a cross-sectional study conducted using interviews with parents visiting outpatient clinics at Saudi Arabia……”
· Line 56-59: Please stay only to the main results of the study (perceptions) exclude the (CI).
· Line 110: add the protocol number
· Table 1 and 4: reform the columns’ and add separate the precedence’s (%)
· Figure 2 and 4: connect the sources of information accordingly: (scientific and health care providers) and the sources (social media and internet).
· Table 4: suggested better statistical analysis in Linkert scale (for example use in two tables analysis as one variable the participants were answered agree and strongly agree against the participants were answered I do not agree and strongly disagree).
Enhance the discussion with the follow references:
· Altulaihi BA, Alaboodi T, Alharbi KG, Alajmi MS, Alkanhal H, Alshehri A. Cureus. Perception of Parents Towards COVID-19 Vaccine for Children in Saudi Population. 2021 Sep 28;13(9):e18342. doi: 10.7759/cureus.18342. eCollection 2021 Sep.
· Szilagyi PG, Shah MD, Delgado JR, Thomas K, Vizueta N, Cui Y, Vangala S, Shetgiri R, Kapteyn A. Parents' Intentions and Perceptions About COVID-19 Vaccination for Their Children: Results From a National Survey. Pediatrics. 2021 Oct;148(4): e2021052335. doi: 10.1542/peds.2021-052335. Epub 2021 Aug 3.
Limitations of the study please add the follow phrase: “This study was conducted at one center in Saudi Arabia and our findings may not present parents' view regarding COVID-19 vaccine across Saudi Arabia”.
Author Response
As suggsted by the respected reviewer all the following modifications have been made
Line 20 affiliations 6-10 please merge in one affiliation
Reply: Done
Line 41 (key words) suggested to authors to reform the key words and exclude the numbers
Reply: Done
Line 55, “In a cross-sectional study conducted using interviews with parents visiting outpatient clinics at King Khalid University Hospital, Riyadh, Kingdom of Saudi Arabia”. Suggested to authors to reform accordingly: “In a cross-sectional study conducted using interviews with parents visiting outpatient clinics at Saudi Arabia……”
Reply: Done
Line 56-59: Please stay only to the main results of the study (perceptions) exclude the (CI).
Reply: Done
Line 110: add the protocol number
Reply: Done
Table 1 and 4: reform the columns’ and add separate the precedence’s (%)
Reply: Table 1 and 4 modified as per instruction.
Figure 2 and 4: connect the sources of information accordingly: (scientific and health care providers) and the sources (social media and internet).
Reply: Figures modified as per instruction.
Table 4: suggested better statistical analysis in Linkert scale (for example use in two tables analysis as one variable the participants were answered agree and strongly agree against the participants were answered I do not agree and strongly disagree).Reply: Table 4 modified as per instruction and constructed separate table 5 for Likert scale .
-
Enhance the discussion with the follow references:
- Altulaihi BA, Alaboodi T, Alharbi KG, Alajmi MS, Alkanhal H, Alshehri A. Cureus. Perception of Parents Towards COVID-19 Vaccine for Children in Saudi Population. 2021 Sep 28;13(9):e18342. doi: 10.7759/cureus.18342. eCollection 2021 Sep.
- Szilagyi PG, Shah MD, Delgado JR, Thomas K, Vizueta N, Cui Y, Vangala S, Shetgiri R, Kapteyn A. Parents' Intentions and Perceptions About COVID-19 Vaccination for Their Children: Results From a National Survey. Pediatrics. 2021 Oct;148(4): e2021052335. doi: 10.1542/peds.2021-052335. Epub 2021 Aug 3.
Reply: Included
Limitations of the study please add the follow phrase: “This study was conducted at one center in Saudi Arabia and our findings may not present parents' view regarding COVID-19 vaccine across Saudi Arabia”.
Reply: Added
Round 2
Reviewer 1 Report
I'm satisfied with responses to my original recommendations and accept the revisions.
Reviewer 2 Report
Thank you, for corrections, very much.
Reviewer 3 Report
Accept in the present form.